# Development of Bilayer Biodegradable Composites Containing Cellulose Nanocrystals with Antioxidant Properties

**DOI:** 10.3390/polym11121945

**Published:** 2019-11-26

**Authors:** Eliezer Velásquez, Adrián Rojas, Constanza Piña, María José Galotto, Carol López de Dicastillo

**Affiliations:** 1Center of Innovation in Packaging (LABEN), University of Santiago de Chile (USACH), Technology Faculty, 9170201 Santiago, Chile; eliezer.velasquez@usach.cl (E.V.); adrian.rojass@usach.cl (A.R.); constanza.pina@usach.cl (C.P.); maria.galotto@usach.cl (M.J.G.); 2CEDENNA (Center for the Development of Nanoscience and Nanotechnology), 9170124 Santiago, Chile

**Keywords:** poly(acid lactic), bilayer, zein, active packaging, quercetin, cellulose nanocrystals

## Abstract

The interest in the development of novel biodegradable composites has increased over last years, and multilayer composites allow the design of materials with functionality and improved properties. In this work, bilayer structures based on a coated zein layer containing quercetin and cellulose nanocrystals (CNC) over an extruded poly(lactic acid) (PLA) layer were developed and characterized. Bilayer composites were successfully obtained and presented a total thickness of approx. 90 µm. The coated zein layer and quercetin gave a yellowish tone to the composites. The incorporation of the zein layer containing CNC decreased the volatile release rate during thermal degradation. Regarding to mechanical properties, bilayer composites presented lower brittleness and greater ductility evidenced by a lower Young’s modulus and higher elongation values. Water permeability values of bilayer composites greatly increased with humidity and the zein coated layer containing quercetin increased this effect. Experimental data of quercetin release kinetics from bilayer structures indicated a higher release for an alcoholic food system, and the incorporation of cellulose nanocrystals did not influence the quercetin diffusion process.

## 1. Introduction

Current food packaging is mainly composed by petroleum-based polymeric materials which present certain disadvantages since, in addition to being synthesized from a non-renewable source, they are not biodegradable polymers, which demonstrates an important source of waste generation and accumulation [1]. Thus, environmental concerns and global awareness of material sustainability have resulted in a strong research interest in the development and use of biodegradable materials [2]. Specifically, poly(lactic acid) (PLA) has become the most commonly used environment-friendly bioplastic because it is commercially available, compostable, and can be processed by conventional melting extrusion. PLA is an aliphatic biopolyester produced from L- and D- lactic acid, which can be derived from fermentation of agricultural products, such as corn or sugarcane [3]. When compared to conventional polyesters, such as polyethylene terephthalate (PET), PLA is still a relatively new material in the plastics industry, but PLA’s demand is increasing due to the reduction of its cost and the favorable regulation worldwide for bio-based materials [4]. PLA basic properties include high mechanical strength and resistance, good and smooth appearance, resistance to chemicals, and others [5,6]. Although some of its properties are attractive, it has limitations such as low impact resistance and relatively low resistance to oxygen and vapor permeabilities when compared to conventional non-degradable polymer resins [7]. Thus, strategies have been developed to improve these properties, such as the addition of nanofillers with remarkable structural and reinforcing properties and/or the development of multilayer structures [8,9,10,11]. In recent years, cellulose nanocrystals (CNC) have been considered one of the most attractive and promising nanoreinforcements for renewable polymers thanks to their unique properties such as their highly crystalline structure, low density, renewable nature, biodegradability, transparency, and high specific area [11,12,13]. This broad interest is related to cellulose properties, such as low cost, chemical modifiability, Young’s high modulus, biodegradability, and abundance in nature, with nanoscale characteristics [14]. On the other hand, multilayer films can achieve specific performance requirements in a cost effective manner in packaging. The superposition of polymeric layers also allows the development of active packaging by incorporating activity to the internal layer that is in contact with the food. Active packaging performs active functions beyond being a passive barrier containing and protecting the food product. It refers to extending the shelf life of foods and improving their safety and quality by means of releasing or removing substances into or from the packaged food or the headspace surrounding the food [15,16,17]. Since oxidation is one of the main food deterioration process, antioxidant active packaging has gained great interest during recent years [18,19]. Multilayer films reduce the necessary amount of the active compound and minimize the losses of the antioxidant to the environment or surrounding by means of the incorporation of the active compound into the internal layer and its release in the food contact area. Although conventional multilayer structures are very common, the development of biodegradable multilayer structures by different techniques, such as thermo-compression assembling and electrospinning coating technology, is quite a recent topic. However, these techniques involve limitations such as high heat exposure that can affect the active layer during compression assembling and small scale production by electrospinning. Specifically, the research on biodegradable bilayer active films has recently emerged [20,21,22].

Thus, in this work, bilayer biodegradable structures based on a thin coated zein layer containing quercetin, as a natural antioxidant compound, and CNC over an extruded PLA substrate were developed with the aim to create an antioxidant biodegradable system with improved physical–mechanical properties for food packaging applications. The novelty of this work lies with the combination of both polymers and the incorporation of quercetin and CNC into the zein inner layer formed by coating, a simple and low-heat exposure technique. CNC can present both effects of improving physical properties of materials and modifying active agent release kinetics.

Zein was used as carrier of antioxidant compounds due to being biodegradable and to its compatibility with quercetin and CNC. This biopolymer is recognized as “GRAS” (Generally Recognized as Safe) by the FDA (U.S. Food and Drug Administration), and it is composed mostly by non-polar amino acids as leucine, proline, alanine, glycine, and valine, which comprises a group of proteins (prolamines) that constitute up to 50–60% of the total protein in the corn’s endosperm [23,24,25]. Moreover, zein films are generally bright and present low permeability to O_2_ and CO_2_ [8,26]. Furthermore, cellulose nanocrystals were incorporated as reinforcement and with the attempt to control the release of the active compound. Thus, the aim of this research was studying the effect of the incorporation of the zein layer containing CNC and quercetin on the structural, optical, thermal, and mechanical properties of PLA-based structures. In addition, the kinetic release of quercetin from active composites was determined.

## 2. Materials and Chemicals

### 2.1. Materials

Poly(lactic acid) (PLA), 2003D (specific gravity ¼ 1.24; MFR g/10min (210 °C, 2.16 kg)), was purchased in pellet form from Natureworks^®^ Co., Minnetonka, MN, USA. Zein (Z 3625) and quercetin (Q) (≥99.5%) was obtained from Sigma-Aldrich (Santiago, Chile). Cellulose nanocrystals (CNC) were purchased from the University of Maine, Orono, ME, USA.

### 2.2. Preparation of Active Bilayer PLA-Zein Composites Structures

#### 2.2.1. Preparation of PLA Film

The PLA film was prepared by using a co-rotating twin-screw extruder Scientific Lab Tech LTE20 (Bangkok, Thailand) at a screw speed of 20 rpm with a temperature profile from 200 to 210 °C. The resulting films presented an average thickness around 80 µm measured by a digital micrometer Mitutoyo ID-C112.

#### 2.2.2. Zein Coating

Three bilayer active films were obtained by coating zein solutions over PLA substrate containing quercetin (Q) at 5 wt % (respect to zein) and three CNC concentrations: 0, 0.5, and 1 wt % (with respect to zein). According to CNC concentration, composites were named PLA/ZN.Q, PLA/ZN.Q.CNC0.5, and PLA/ZN.Q.CNC1, respectively. First, 0.225 g of quercetin was dissolved in 24 mL of ethanol and cellulose nanocrystals were dispersed in 6 mL of water by sonication during 10 min in a Fisher Scientific sonicator (Model CL-334, pulse 15 and amplitude of 90%). Both solutions were mixed and 4.5 g of zein was added and stirred at room temperature for 1 h. Control films without quercetin (PLA/ZN.CNC) were also prepared following similar methodology in order to study the effect of the polymer and the nanocrystals. Before the zein coating process, a corona treatment on PLA film was carried out by using a FT-800*350mm equipment to improve zein adherence. Coatings were performed with a RK-Print multicoater equipment model K303 by applying 3.5 mL of each solution onto PLA films using a stainless steel rod No. 5 at 5 m min^−1^. Finally, coatings were dried by application of hot air for 50 s and their thickness values were approximately 9–10 µm.

### 2.3. Characterization of Optical and Structural Properties

#### 2.3.1. Optical Characterization of Bilayer Structures

The CIELAB chromaticity coordinates, lightness *L**, *a** (red to green tones), and *b** (blue to yellow tones) parameters were measured through a CR-410 Minolta chromameter colorimeter (Konica Minolta, Santiago, Chile) using a white standard color plate, D65 illuminant, and 2° observer (*L** = 97.77, *a** = −0.03, *b** = 1.94). Color parameters was the average from six measurements along each film. The color difference (Δ*E**) of each sample was calculated respect to PLA film by Equation (1):Δ*E** = [(Δ*L**)^2^ + (Δ*a**)^2^ + (Δ*b**)^2^]^1/2^(1)

#### 2.3.2. Morphological Characterization of Bilayer Structures

Bilayer structures were analysed by Optical Microscopy using a Zeiss Standard 25 ICS microscope (Zeiss, Oberkochen, Germany). Materials were stained with lugol in order to observe zein coating due to its property to link with protein molecules.

#### 2.3.3. Fourier Transform Infrared (FTIR)–Attenuated Total Reflectance (ATR) Spectroscopy

FTIR–ATR spectroscopy was used to characterize the presence of specific chemical groups in the materials. FTIR spectra were performed in ATR mode with a Bruker IFS 66V spectrometer (Bruker, Ettlinger, Karlsruche, Germany). The spectra were the results of 64 co-added interferograms at 4 cm^−1^ and resolutions in the wavenumber range from 4000 to 400 cm^−1^. The spectra analyses were performed using OPUS Software Version 7 (Bruker, Ettlinger, Karlsruche, Germany).

### 2.4. Thermal Properties

#### 2.4.1. Differential Scanning Calorimetry

Differential Scanning Calorimetry (DSC) analyses were performed in a Mettler Toledo DSC-822e calorimeter (Mettler Toledo, Schwerzenbach, Switzerland). Amounts of 8–10 mg of the samples were heated from 25 to 200 °C at a heating rate of 10 °C min^−1^ under a nitrogen atmosphere. Glass transition temperature (*T*_g_), cold crystallization (*T*_cc_), and melting (*T*_m_) temperatures, as well as cold crystallization (Δ*H*_cc_) and melting (Δ*H*_m_) enthalpies were determined.

#### 2.4.2. Thermogravimetry Analysis

Thermogravimetric analyses (TGA) of films were carried out using a Mettler Toledo Gas Controller GC20 Stare System TGA/DSC (Mettler Toledo, Schwerzenbach, Switzerland). Samples were heated from 30 to 700 °C at 10 °C min^−1^ under nitrogen atmosphere with a flow rate of 50 mL min^−1^. Onset decomposition temperature (*T*_onset_) at *T*_15%_ and maximum degradation rate were determined by the STARe software (Mettler Toledo, Schwerzenbach, Switzerland). Volatile release rate was calculated as the weight loss between *T*_onset_ and *T*_d_,_max_ divided by the temperature difference *T*_d,max_–*T*_onset_.

### 2.5. Mechanical Properties

The mechanical properties were determined using a Zwick Roell Tensile Tester (model BDO-FB 0.5 TH, Ulm, Germany), according to ASTM D-882. Strips (21 cm × 2.5 cm) were conditioned at 23 °C and 50% HR during 48 h, and analyzed with a 1 kN load cell. The initial grip separation was 12.5 cm and the crosshead speed used was 12.5 mm min^−1^. Young’s modulus, tensile strength, and elongation at break were reported as the average of ten replicates.

### 2.6. Water Vapor Permeability (WVP) Analysis

WVP tests were carried out at 50 and 90% relative humidity (RH) and 37 °C using aluminum permeability cups in accordance with standard method of ASTM E96. The aluminum cups were filled with 7 g of silica gel and sealed with vacuum silicon grease and the film to be tested. The film was fixed in place with a flat Viton ring, an aluminum ring, and three press-screws. To ensure the necessary relative humidity, the cups were then stored in desiccators containing salt solutions: magnesium nitrate (Mg(NO_3_)_2_) and potassium sulfate (K_2_SO_4_) for 50 and 95% RH, respectively. The cups were daily weighed, and the plot of the weight increment versus time provided the water vapor transmission rate. These values were then divided by the water pressure gradient and film area and multiplied by the sample thickness to obtain the WVP value.

### 2.7. Study of Release Kinetics of Quercetin from Active Bilayers Composites

#### 2.7.1. Release Assay Procedure

Quercetin release kinetics have been characterized by means of specific experimental migration assays using 3% (*v/v*) acetic acid and 10% (*v/v*) ethanol as food simulants in order to describe the mass transfer of quercetin from bilayer PLA/ZN composites. Migration experiments were carried out in accordance with the European Committee for Standardization [27,28,29]. PLA/ZN composites containing quercetin were totally immersed into glass tubes filled with the food simulants in a relation of 6 dm^2^ L^−1^. A release assay was carried out at 40 °C and quercetin released was periodically measured by UV spectroscopy at 425 nm using a UV-VIS Spectrophotometer Pharo 300 Spectroquant^®^ [30]. Release assays were carried out until the equilibrium condition was reached, when quercetin released was maintained constant in time in at least two continuous consecutive measurements.

#### 2.7.2. Determination of Partition and Diffusion Coefficients of Quercetin in PLA/ZN Bilayers

Quercetin release from bilayer PLA/ZN composites were described by means of an effective Fick’s law approach. Thus, the mass transfer process for the analyzed systems can be explained by means of a resistances-in-series approach based on the one-dimensional simplification of Fick’s Law [27,31,32]. Therefore, instantaneous mass transfer of quercetin can be estimated by molecular diffusion through the materials. This mechanism can be described through Equation (2):(2)$$J_{I}=\frac{D_{eff}}{L/2}\times C_{Quer}^{P}\left(x=0,t\right)-C_{Quer}^{P}(x=L/2,t)$$
where *J*_I_ is the mass transfer flux (kg m^−2^ s^−1^) of the released quercetin through the PLA/ZN sample, *D_eff_* is the effective diffusion coefficient of quercetin in the materials (m^2^ s^−1^), $C_{Quer}^{P}$(kg m^−3^) is the concentration of quercetin in the material bulk, and *L* is the film thickness (m). In this case, the configuration of the release experiments allows the symmetrical description of the mass transfer with Equation (2) from both sides of the sample. The dimensionless distribution coefficient of quercetin between the PLA/ZN samples and the food simulant is represented by the ratio of concentrations in the interface through Equation (3):(3)$$K_{P/FS\;}=\;\frac{C_{Quer}^{p}\;(x=L/2,t)}{C_{Quer}^{FS}\;(x=\;L/2,t)}$$

This value can be directly estimated from the results of the release experiments at the equilibrium condition, where $C_{Quer}^{FS}\;(x=L/2,t)$ is the equilibrium concentration of quercetin at the interface in the food simulant and $C_{Quer}^{P}\;(x=L/2,t)$ is its value at the interface in the PLA/ZN sample. This last value can be estimated as a function of time through mass balance.

Finally, the next step in the release process is the transfer through the boundary layer in the food simulant phase at the proximities of the interface. The transfer mechanism considered in the model was natural convection because the food simulant solutions were not stirred during the release assays. Thus, this step can be described by Equation (4):(4)$$J_{II}=k\cdot (C_{Quer}^{FS}\;(x=\;L/2,t)-C_{Quer}^{FS}\left(x=\infty ,t\right)$$
where $C_{Quer}^{FS}$ is the quercetin concentration in the food simulant bulk (kg m^−3^) and k (m s^−1^) is the quercetin mass transfer coefficient that quantifies natural convection in the food liquid phase. Its value was calculated by means of the correlation reported by Galotto and coworkers, where the coefficient is obtained from the Sherwood number, which is calculated as a function of the Grashof and Schmidt numbers [27]. A few assumptions had to be made in order to use this model:

1. The initial quercetin concentration in the materials is known and is homogeneously distributed, $C_{Quer}^{p}\left(x=0,t\right)=C_{Quer,0}^{P}=0\leq x\leq L/2$.

2. The food simulant is initially curcumin-free: $C_{Quer}^{FS}\;\left(x,0\right)=0$.

3. There is no chemical interaction between food simulant and polymer (no swelling).

4. There is no mass transfer limitation in the liquid food simulant, so curcumin is always homogeneously distributed in the food simulant.

These mathematical model equations describing molecular diffusion of quercetin through the materials, as well as the polymer/food simulant distribution coefficient and transfer through the boundary layer in the food simulant phase at the proximities of the material/food interface were applied in a previous study developed by the same working group [27,28,29]. Mass transfer equations were solved under steady-state condition for an instantaneous time. The Regula Falsi method was applied in order to reduce the number of interfacial concentration values of curcumin iterations. Taking into account that the initial quercetin concentration in the PLA/ZN samples and the simulant bulk are known, an iterative calculation can be done to estimate the mass transfer flux through the interface at steady-state condition (*J*_I_ = *J*_II_) for a specific time step. Iterative variables in this calculation are the interfacial concentrations of quercetin in both phases. When this numerical solution was identified, the bulk concentrations of quercetin in the polymer and in the food simulant were recalculated by mass balance, starting a new iterative calculation for the next time step. These calculations were implemented by means of a program built in Matlab 7.1 (MathWorks, Natick, MA, USA). Simulations were developed, using experimental values of the *K*_P/FS_ defined in Equation (3). Different Deff values of quercetin in the materials were considered involving the lowest values of RMSE (root of the mean-square error) between experimental data and values predicted by the mathematical model. The root of the mean-square error (RMSE (%)) was calculated using Equation (5). This measures the fit between experimental and estimated data [27]:(5)$$RMSE=\frac{1}{M_{M,0}}\cdot \sqrt{\left(\frac{1}{N}\right)\cdot \sum_{i=1}^{N}{\left({\left(M_{FS,t}\right)}_{experimental,i}-{\left(M_{FS,t}\right)}_{predicted,i}\right)}^{2}}$$
where *N* is the number of experimental points for each release curve; *i* is the observations number; *M*_M,0_ is the initial amount of quercetin in the material (µg); and *M*_FS,t_ is the quercetin amount in the food simulant at time *t* (µg).

### 2.8. Statistical Analysis

The experimental design was random-type. Data analysis was carried out using Statgraphics Plus 5.1 (StatPoint Inc., Herndon, VA, USA). This software was used to implement variance analysis and Fisher’s LSD test. Differences were considered significant at *p* < 0.05.

## 3. Results and Discussion

### 3.1. Optical Properties

Color parameters of developed biodegradable composites are presented in Table 1. Lightness values (*L**) significantly decreased in zein-coated PLA including quercetin and CNC. Significant changes occurred when PLA film was coated with zein: (i) *a** parameter, that moves from red tones (+*a**) to green tones (−*a**), shifted towards more negative values; and (ii) an increase in *b** parameter, which indicated variations from blue tones (−*b**) towards yellow tones (+*b**). This effect on the chromaticity coordinates *a** and *b**, and therefore on the global color variation, was significantly increased with the addition of quercetin. This antioxidant compound is a flavonoid and a natural pigment greatly characterized by its intense yellow color [30]. The incorporation of CNC into the zein coating, principally at 1 wt %, intensified this effect without lightness changes. Nevertheless, the incorporation of CNC in the absence of quercetin did not entail significant effect on the color variation.

### 3.2. Morphological Results

Optical microscopies of active bilayer structures PLA/ZN.Q, PLA/ZN.Q.CNC0.5, and PLA/ZN.Q.CNC1 shown in Figure 1 confirmed a good physical adhesion between PLA and zein coating. The coated zein layer with the presence of lugol acquired a darker tone due to the reaction between this compound with zein proteins. As it was expected, the thicknesses of both layers in each sample were representative of the average thicknesses (around 80 µm and 9–10 µm for PLA and zein, respectively).

### 3.3. FTIR Spectra Results

Figure 2 shows the FTIR spectra of control PLA and bilayer PLA composites containing a coated zein layer. Monolayer PLAs presented their characteristic peaks, such as the carbonyl vibration at approximately 1747 cm^−1^, and peaks at 1040, 1080, and 1180 cm^−1^, associated with C–O stretching, C=O, and C–O symmetric stretching and C–O–C stretching, respectively [33]. Nevertheless, because FTIR–ATR technique only analyses the first microns of the material, the resulting spectra of bilayer composites presented principally zein characteristic bands, and PLA structure was completely hidden. In addition to the fact quercetin and cellulose nanocrystals were incorporated at low concentrations, all coated composites presented similar spectra corresponding to zein characteristic bands. The main zein peaks were correlated to zein amides I, II, and III, respectively: (i) peak at 1647 cm^−1^ associated to C=O stretching vibration band of peptide groups with C–N stretching contributions; (ii) in the range of 1538 cm^−1^, characterized by vibrations in the plane of N–H bond of peptide groups; and (iii) in the region of 1242 cm^−1^ and a set of band between 3100 and 2800 cm^−1^, associated with C–H vibrations of CH_3_ and CH_2_ groups from fatty acids and side chains from amino acids [34]. Zein peaks at 1041–1237 cm^−1^ corresponded to C–N stretching and hydroxyl groups were also observed through a broad band with a peak at approximately 3288 cm^−1^ [35]. The incorporation of quercetin was only evidenced with the appearance of a weak peak at 1199 cm^−1^ associated to C–O stretching in phenol.

Chemical interactions, principally through hydrogen bonding-type, between this antioxidant compound and cellulose nanocrystals, were shown through a slight displacements of quercetin characteristic peak to lower values, at approximately 1164 cm^−1^ (see Figure 2B). The cellulose nanocrystal bands were probably overlapped with zein spectra. A new band appeared at approximately 1643 cm^−1^ and is commonly associated to adsorbed water in cellulose [36,37]. Slight chemical interactions between zein and CNC was evidenced by short displacements of zein band from 1175 to 1167 cm^−1^ associated to C–N stretching of amine groups [35]. Chemical interactions between zein, quercetin, and CNC could be occurring through hydrogen bonding.

### 3.4. Thermal Characterization

#### 3.4.1. Differential Scanning Calorimetry

Thermal parameters obtained during the heating process are shown in Table 2. Control PLA film showed a glass transition around 64 °C and a crystal rearrangement at 117.8 °C. An endothermic transition with double melting peaks at 149.4 and 154.4 °C, characteristic peaks of this type of semicrystalline polymers, was detected due to the presence of crystals with different morphologies or lamellar thicknesses formed during crystallization of the material and cold rearrangement [38,39].

Statistical analyses were not presented because all thermal parameters values did not show significant differences according to ANOVA analysis (*p* < 0.05).

In general, thermal parameters of all zein-coated PLA films were statistically similar to neat PLA. This fact can be attributed because, instead of strong interactions, an interfacial physical adhesion between PLA and the thin zein coating containing Q and CNC occurred. In addition, the absence of thermal transitions associated with the components in the coating (ZN, Q, and CNC) was due to their low concentrations and a PLA melting temperature close to zein glass transition temperature *T*_g_ at approximately 159 °C [40].

#### 3.4.2. Thermogravimetric Analysis

Thermogravimetric analysis was also carried out to observe degradation processes and the thermal stability of developed bilayer composites. Table 3 presents the onset decomposition and degradation temperatures, weight loss, and the volatile release rate at this temperature range, and Figure 2 shows curves of weight loss of composites and their derivatives.

The zein polymer presented a water removal between 42 and 114 °C, followed by the evaporation of fatty acids or lightly stable amino-acids, and finally the maximum degradation of protein chains that occurred between 265 and 400 °C [41], as it is shown in Figure 3. The thermal stability of PLA was slightly decreased by the incorporation of zein coating due probably to the earlier degradation of this protein when compared to PLA. This fact was evidenced by a decrease in the onset and maximum degradation temperatures of PLA/ZN bilayer film (Table 3). On the other hand, zein coating caused a barrier effect evidenced by a decrease of volatile release rate, considering that the weight loss rate of zein is much lower (0.3 wt %/°C) than pure PLA.

TGA thermograms of Q and CNC were also analyzed and the maximum degradation rate of the quercetin and cellulose nanocrystals occurred at the typical temperatures of these components, 355 and 292 °C, respectively [42,43]. Two degradation stages were observed for anhydrous quercetin, while CNC analysis evidenced a degradation attributed to the removal of hydroxyl and –CH_2_–OH groups starting from 289 °C and the decomposition of cellulose chains at higher temperatures (see Figure 3A) [43,44,45]. The weight loss of CNC up to 125 °C was associated with the volatilization of water. DTGA thermograms of bilayer structures revealed a principal peak associated with the degradation of all components from the bilayer material (Figure 3B). The incorporation of CNC and quercetin did not affect *T*_onset_ and *T*_d,max_ compared to the PLA/ZN control values (see Table 3). A reduction of the volatile release rate was observed with the incorporation of ZN.Q.CNC layer, showing weight losses between 44 and 47 wt %, as well as a slight increase in the *T*_onset_ and *T*_d_ with respect to the PLA/ZN control. This fact can be attributed to possible interactions between zein polymer, cellulose nanocrystals, and quercetin. The interactions could be (i) non-covalent hydrophobic and hydrogen-bonding between the quercetin phenolic structure and protein [24]; and (ii) hydrogen bonds between ZN-Q, CNC-ZN, and CNC-Q [46]. A strong network resulted from hydrogen bonding-type interactions between quercetin hydroxyl groups, cellulose hydroxyls, and zein amine and carbonyl groups, additionally quercetin and CNC oxygen (ether groups) could be involved in this type of interaction.

### 3.5. Mechanical Properties

Bilayer PLA/ZN films revealed a decrease in the elastic modulus and the maximum force supported before irreversible deformation, as well as an increase of approximately 65% in elongation at break compared to monolayer PLA control film due to a plasticizing effect produced by zein coating incorporation (Table 4). The good physical adhesion of the zein layer onto PLA verified by optical microscopy implied the development of a material with lower brittleness and greater ductility evidenced with a lower Young’s modulus and higher elongation. This effect is more significant in the presence of quercetin and CNC at 1 wt %. These changes represent a relevant improvement considering that low extendibility, stiffness, and brittleness of PLA could cause problems during packaging transportation and limited its use for some applications [47]. The incorporation of quercetin and/or CNC in the zein coating produced a tendency: (i) to increase Young’s modulus (stiffness); (ii) to increase ductility, evidenced by higher elongation values; and (iii) to decrease the maximum resistance with respect to the PLA/ZN film, as shown in Table 4. These facts could be due to the formation of the aforementioned network based on hydrogen-bonding interactions between zein, Q, and/or CNC, as it was observed by TGA analysis, which increased the capacity of the film to deform at lower tensile strength. This effect was statistically significant only for the PLA/ZN.Q. CNC1 sample, attributed to the highest concentration of cellulose nanocrystals.

### 3.6. Barrier Properties

Water vapor permeability results of PLA-based composites at both relative humidity conditions are gathered in Figure 4. Due to the hydrophilic nature of both polymers, all films presented the same trend, permeability values significantly increased with the RH gradient, showing the plasticizing effect of water in both PLA and zein polymers. The sorption of water increased the mobility of polymeric chains and, consequently, water diffusivity values enhanced.

The incorporation of the zein layer did not significantly alter the water barrier of PLA. Although zein is a relatively hydrophobic protein, this biopolymer was highly affected and strongly plasticized by water. Permeability values of PLA are relatively low when compared to other bioplastics, such as polycaprolactone, cellulose acetate, and cellulose acetate propionate [48]. Thus, the improvement of the vapor barrier at high RH was not achieved. At increasing RH conditions, PLA unique layer presented lower enhancement of vapor permeability tan bilayer structures. The reason can be related to the fact that PLA does not take up water to a significant extent when compared to zein. In this bilayer system, PLA is the highest barrier and the dominant phase determining the final barrier. Thus, the incorporation of quercetin and cellulose nanocrystals into the zein layer did not modify significantly barrier values, as it was expected. The differences on PLA and zein thickness also influenced on lowering the protection that CNC could perform. Moreover, CNC dispersion was probably not efficient enough to afford a barrier effect. In general, the addition of this zein layer implied a slight negative effect on the water permeability of the bilayer system due to the higher free volume attained and the corresponding changes in diffusion and also in solubility of water of the zein layer. When compared to its oil-based counterpart PET, which has a WVP of 3 × 10^−15^ (Kg m/s m^2^ Pa) [49], it is worthy to highlight values are not so badly enhanced.

### 3.7. Kinetic Release of Quercetin from Bilayer Composites

Figure 5 shows the quercetin release kinetics, expressed as mg of quercetin per surface, from developed bilayer PLA/ZN samples into 3% acetic acid (*v/v*) and 10% ethanol (*v/v*) as food simulants. The graphs represent experimental and theoretical estimations, which were obtained through the mass transfer model with correlated diffusion coefficients of quercetin in the bilayer samples. Quercetin release kinetics were characterized by means of specific experimental assays in order to describe the mass transfer of quercetin from PLA/ZN bilayer samples. Thus, the distribution coefficient of quercetin at the interphase between the materials and food simulants (*K*_P/FS_) and the diffusion coefficient of quercetin in the different PLA/ZN samples (Dp) were calculated. *K*_P/FS_ was estimated when the released quercetin reached the plateau during each release kinetics profile.

Table 5 shows partition coefficient (*K*_P/FS_) and effective diffusion coefficient (*D*_eff_) values of quercetin from different PLA/ZN composites. In addition, the value of root mean square error (RMSE) of the model solution related to experimental data was also reported.

PLA/ZN bilayer samples presented similar *K*_P/FS_ values because all the samples present the same initial amount of quercetin, and this behavior was observed for both food simulants. *K*_P/FS_ values of composites into 10% EtOH food simulant were slightly lower than values obtained on 3% acetic acid because the highest affinity of quercetin towards ethanol. Regarding the effect of the incorporation of cellulose nanocrystals (CNC), results evidenced the addition of CNC into coated zein layer did not play any role in the cohesive energy established between quercetin and polymer. As release kinetic curves in Figure 5 and RMSE values in Table 5 show, theoretical curves greatly fitted with the experimental data, showing that release of quercetin from bilayer PLA/ZN composites was principally controlled by the diffusion phenomenon in the zein layer defined by the second Fick’s law. This behavior was in agreement with the release of other active agents from zein layer and films [50,51,52]. The curve slope of quercetin kinetic releases depended on the presence of CNC in the zein layer and on the type of food simulant. Table 4 also presents the values of the diffusion coefficient of quercetin through bilayer PLA/ZN.Q, PLA/ZN.Q.CNC0.5, and PLA/ZN.Q.CNC1 composites in 10% EtOH and 3% acetic acid. These values ranged between 1.5 and 4.5 × 10^−15^ m^2^ s^−1^ and were lower than the diffusion coefficient value reported for quercetin in PLA monolayer (6.4 × 10^−15^ m^2^ s^−1^), showing that the incorporation of quercetin into a coated zein layer over PLA allowed a more sustained release [53]. These results are very promissory for the development of active food packaging materials of sustained release. Diffusion coefficients of quercetin in PLA/ZN bilayers slightly increased as CNC concentration in the zein layer increased, and this trend was observed for both food simulants. The estimated diffusion coefficient of quercetin from the PLA/ZN.Q.CNC1 composite on 10% EtOH food simulant was 3.0 × 10^−15^ m^2^ s^−1^. This value was twice the estimated value for the bilayer PLA/ZN.Q composite without CNC (1.5 × 10^−15^ m^2^ s^−1^) and this fact can be explained due to the low dispersion of CNC inside the coated zein layer that caused a possible lack of cohesion in the zein matrix, which improved quercetin mass transfer during release assays. The inefficient CNC dispersion through the zein polymer was already supported by water vapor permeability results. Diffusion coefficients of quercetin in 3% acetic acid were slightly higher than values obtained in 10% EtOH. These results could be explained in terms of zein plasticization and swelling due to the penetration of the food simulant that modified the effective transport properties of zein during the release assay, improving quercetin mass transfer and overestimating the value of the diffusion coefficient of quercetin. To our knowledge, any previous data on quercetin diffusion coefficient from zein coated polymers or another bilayer has been reported, but as a reference point, this value was lower than the diffusion coefficient of thymol (3.8 × 10^−14^ m^2^ s^−1^) and carvacrol (4.9 × 10^−14^ m^2^ s^−1^) in zein films using 10% *v/v* EtOH as food simulant [54]. This result was expected because diffusion coefficient magnitude decreased as the molecular weight of the solute increased.

## Figures and Tables

**Figure 1 polymers-11-01945-f001:**
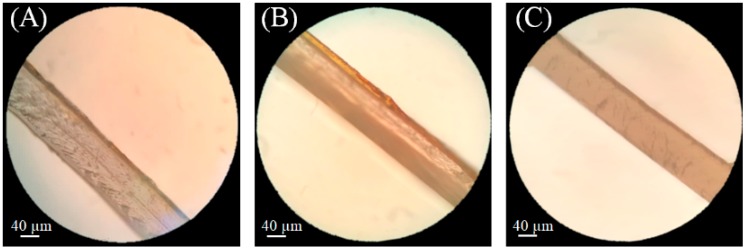
Optical micrographs of: (**A**) PLA/ZN, (**B**) PLA/ZN.Q.CNC0.5, and (**C**) PLA/ZN.Q.CNC1.

**Figure 2 polymers-11-01945-f002:**
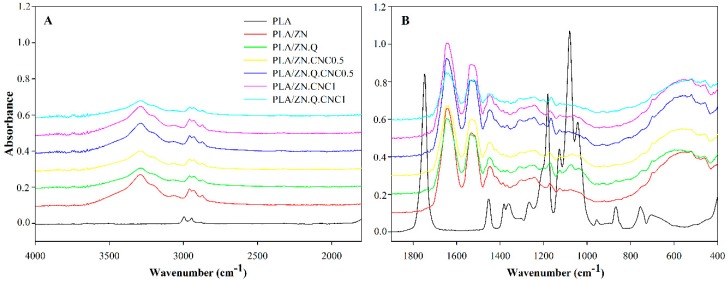
(**A**) Fourier-transform Infrared Spectroscopy (FTIR) spectra of developed PLA composites; (**B**) with close up to lower wavenumbers.

**Figure 3 polymers-11-01945-f003:**
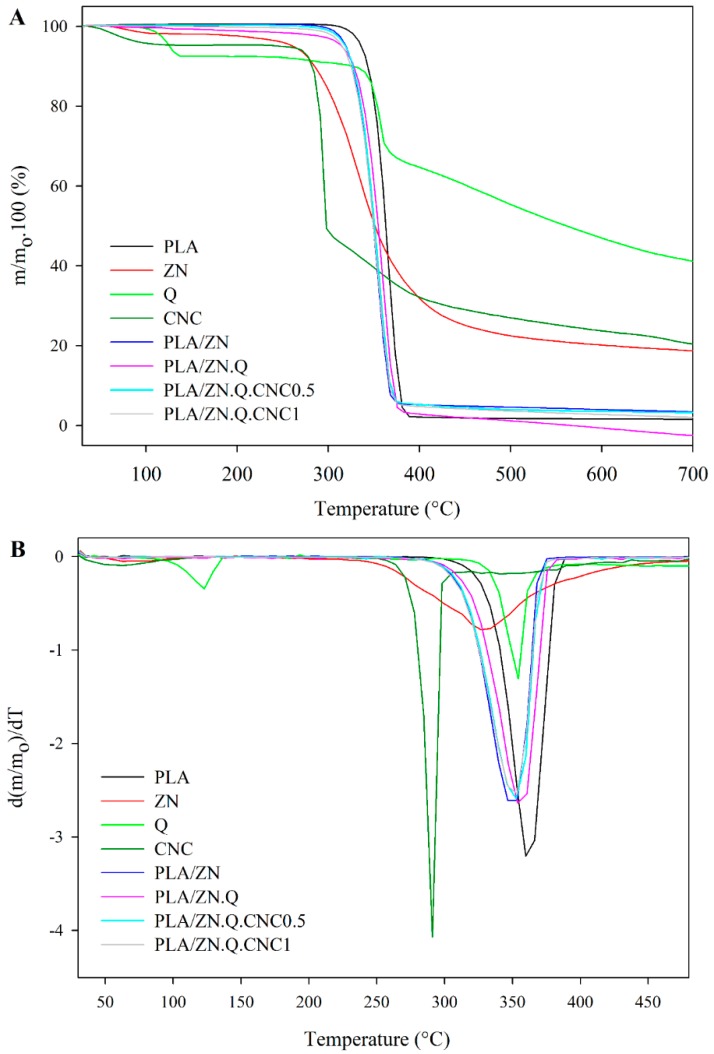
TGA (**A**) and their derivative curves DTGA (**B**) curves of the components and films.

**Figure 4 polymers-11-01945-f004:**
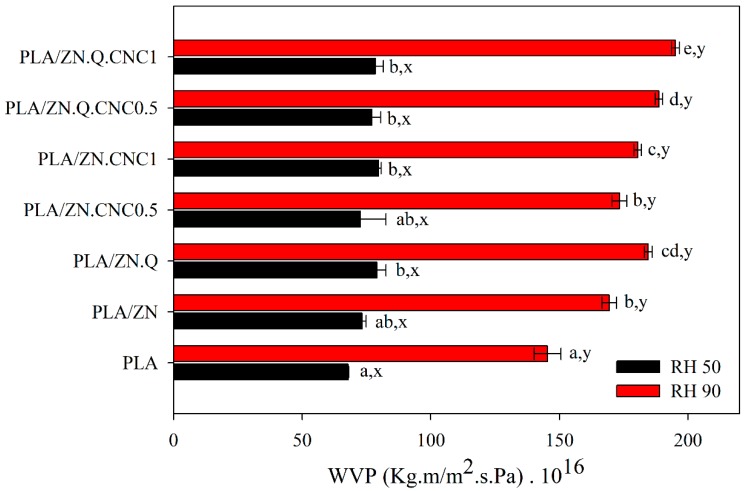
Water vapor permeability values for PLA-based materials (case letters a–e indicate significant differences among the values of permeability of different films at the same RH; letters x,y indicate significant differences among the values of permeability of the same sample at different values of RH).

**Figure 5 polymers-11-01945-f005:**
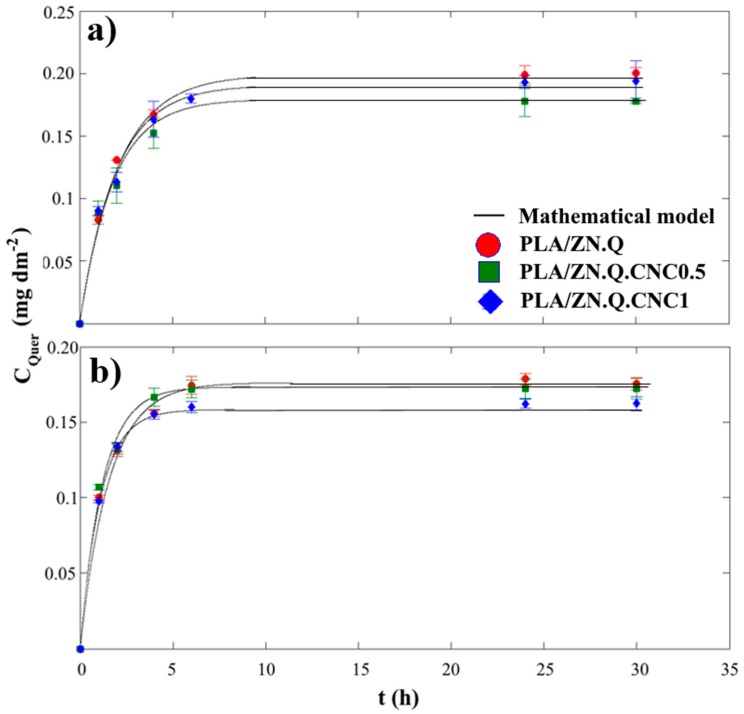
Release kinetic of quercetin from PLA/ZN to food simulants: (**a**) 10% (*v/v*) ethanol and (**b**) 3% acetic acid.

**Table 1 polymers-11-01945-t001:** Chromaticity coordinates and color variation of the composites.

Films	*L**	*a**	*b**	Δ*E*
PLA	98.3 ± 0.2 ^d^	−0.06 ± 0.01 ^d^	2.27 ± 0.02 ^a^	-
PLA/ZN	97.9 ± 0.2 ^bc^	−0.57 ± 0.03 ^c^	4.10 ± 0.11 ^b^	1.95 ± 0.15 ^a^
PLA/ZN.CNC0.5	98.0 ± 0.1 ^c^	−0.60 ± 0.04 ^c^	4.24 ± 0.16 ^b^	2.07 ± 0.17 ^a^
PLA/ZN.CNC1	98.0 ± 0.2 ^c^	−0.60 ± 0.02 ^c^	4.25 ± 0.04 ^b^	2.08 ± 0.06 ^a^
PLA/ZN.Q	97.8 ± 0.2 ^ab^	−1.60 ± 0.23 ^b^	6.65 ± 0.71 ^c^	4.68 ± 0.75 ^b^
PLA/ZN.Q.CNC0.5	97.7 ± 0.2 ^a^	−1.67 ± 0.08 ^b^	6.85 ± 0.22 ^c^	4.90 ± 0.26 ^b^
PLA/ZN.Q.CNC1	97.7 ± 0.1 ^a^	−1.81 ± 0.07 ^a^	7.28 ± 0.22 ^d^	5.36 ± 0.24 ^c^

Lower case letters a-d indicate statistically significant differences of the same parameter among films according to ANOVA analysis (*p* < 0.05).

**Table 2 polymers-11-01945-t002:** Differential Scanning Calorimetry (DSC) parameters and crystallinity of developed composites.

Films	*T*_g_ (°C)	*T*_cc_ (°C)	Δ*H*_cc_ (J g^−1^)	*T*_m1_ (°C)	*T*_m2_ (°C)	Δ*H*_m_ (J g^−1^)
PLA	64.2 ± 0.8	117.8 ± 0.1	18.2 ± 8.2	149.4 ± 0.5	154.4 ± 0.9	20.2 ± 8.8
PLA/ZN	63.7 ± 0.6	118.1 ± 0.2	24.1 ± 0.6	150.3 ± 0.1	153.6 ± 1.0	25.7 ± 0.3
PLA/ZN.Q.CNC0.5	62.8 ± 3.9	115.4 ± 0.7	20.5 ± 0.6	147.7 ± 2.6	154.6 ± 0.4	22.7 ± 0.1
PLA/ZN.Q.CNC1	65.4 ± 0.2	115.6 ± 1.5	23.9 ± 1.7	149.3 ± 0.2	154.2 ± 0.4	25.4 ± 1.6

**Table 3 polymers-11-01945-t003:** Onset and degradation temperatures, weight loss and volatile release rate during thermogravimetric analysis (TGA).

Films	*T* _onset_	*T* _d,max_	Weight Loss between *T*_onset_ and *T*_d,max_ (%)	Volatile Release Rate (wt %/°C)
PLA	348.0	365.9	44.0	2.5
PLA/ZN	330.5	353.3	45.9	2.0
PLA/ZN.Q.CNC0.5	330.6	354.8	46.9	1.9
PLA/ZN.Q.CNC1	331.0	354.0	44.7	1.9

*T*_onset_: onset decomposition temperature. *T*_d,max_: temperature at maximum degradation rate.

**Table 4 polymers-11-01945-t004:** Tensile parameters of PLA composites.

Films	Young’s Modulus (MPa)	Maximum Resistance (MPa)	Elongation at Break (%)
PLA	2768 ± 154 ^b^	58.7 ± 2.0 ^d^	6.1 ± 2.3 ^a^
PLA/ZN	2410 ± 393 ^a^	50.5 ± 3.4 ^c^	10.1 ± 5.3 ^b^
PLA/ZN.Q	2627 ± 138 ^ab^	48.1 ± 2.1 ^b,c^	11.8 ± 2.7 ^b,c^
PLA/ZN.CNC0.5	2602 ± 250 ^a,b^	49.2 ± 3.2 ^b,c^	11.6 ± 4.2 ^b,c^
PLA/ZN.CNC1	2566 ± 355 ^a,b^	48.7 ± 2.7 ^b,c^	12.1 ± 3.2 ^b,c^
PLA/ZN.Q.CNC0.5	2631 ± 75 ^a,b^	48.1 ± 2.7 ^b^	11.3 ± 3.1 ^b,c^
PLA/ZN.Q.CNC1	2453 ± 300 ^a^	44.8 ± 2.0 ^a^	14.5 ± 4.7 ^c^

Lower case letters a–c indicate statistically significant differences of the same mechanical parameter among films according to ANOVA analysis (*p* < 0.05).

**Table 5 polymers-11-01945-t005:** Partition and diffusion coefficients and root of the mean-square-error (RMSE) values of quercetin from PLA/ZN samples into different food simulants at 40 °C.

Simulant	Bilayer PLA/ZN Composite	*K* _P.FS_	Dp (m^2^ s^−1^)	RMSE
10% EtOH	PLA/ZN.Q	5422 ± 139	1.5 × 10^−15^	1.20
	PLA/ZN.Q.CNC0.5	5353 ± 45	2.0 × 10^−15^	1.05
	PLA/ZN.Q.CNC1	5869 ± 103	3.0 × 10^−15^	0.43
3% acetic acid	PLA/ZN.Q	6330 ± 220	2.0 × 10^−15^	0.67
	PLA/ZN.Q.CNC0.5	5869 ± 150	3.5 × 10^−15^	0.34
	PLA/ZN.Q.CNC1	6171 ± 150	4.5 × 10^−15^	0.71

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
