# Peer review of "Development of Bilayer Biodegradable Composites Containing Cellulose Nanocrystals with Antioxidant Properties"

_polymers, 2019, doi:10.3390/polym11121945_

Round 1

Reviewer 1 Report

This paper reports on the preparation and characterization of a new packaging with a bi-layered structured made up of a fist PLA layer and a second zein layer containing quercetin and cellulose nanocrystals. PLA packaging are known to be brittle, as explained in the paper, and many other papers describes the improvement of mechanical properties of PLA. The use of multiple-layered packaging is an approach to overcome this issue.

The paper is easy to follow, and the characterizations are convincing. The packaging reported in this paper is obviously new but is it original? In other words, they are so many polymer materials used in packaging application and it is impossible to publish all of them in a scientific journal. In a journal such as Polymers, I expect to see a new concept, a new approach, a clear technological improvement useful to other readers, a better understanding of synthesis, preparation and properties of polymer material. By considering these criteria and after a careful reading of the paper, it is still difficult for me to find the nature of the original message that the authors desire to transmit to the readership of polymers. It is important to answer to that question to avoid publishing only technical papers with new material but with no scientific originality. I was searching some answers in the conclusion but no one was provided by the author at the end of the paper.

It is possible at the problem stems just from the form of the paper and I recommend thus a major revision, which gives the opportunity to the authors to answer to my criticism and update accordingly their paper.

Author Response

The authors understand the reviewer's argument. In this sense, authors consider that this work contains scientific originality and has generated results that are worthy of publishing to share this knowledge with the scientific and technological society. In general, this work did not simply meant the development of a polymeric coating over a polymer substrate. Hereby, authors have studied the development of an active material differently from many previously developed studies where an active material had been developed only through the addition of an active agent in a polymer and the characterization was proceeded. In this case, the proposed system consisted on a bilayer structure where the inner layer serves as a carrier of the active compound. Thus, the effect of the incorporation of this second layer on thermal, structural, mechanical and barrier properties was studied. On the other hand, the effect of the addition of cellulose nanocrystals and the active agent on the bilayer structure properties was also studied. The third and most interesting point was the study of the effect of different simulants and the addition of these nanofillers on the kinetic release processes of this antioxidant agent.

Sentences were included in Introduction section and discussion of results, principally regarding kinetics release, in order to maximize the scientific importance of this study.

Reviewer 2 Report

Dr. Velasquez and co-workers report a very interesting work of using quercetin (Q) as active component of zein (Z)-cellulose nanocrystals (CNC) as coating for poly(lactic acid) (PLA) film. The obtained bilayer systems are intended for active food packaging. The topic is interest ad well related with the scope of the journal. The work is well organized and the materials were fully characterized, while the understanding of release process is very interesting. Therefore, I would like to recommend this work to be published in Polymers after minor revision to address some of my comments below:

-Line 31: poly (lactic acid) (PLA) should be poly(lactic acid) (PLA) (the space in polymer names should be avoided).

-Lines 37-39: Please add some references.

-Lines 55-56: some more recent references than 2014 from the last years (last two or three years) should be included.

- Lines 64-64: some references should be added and discussed to justify the interest as well as the recent novelty of the topic suggested by the authors.

- I suggest authors to describe better which is the aim of the work. Why authors decided to incorporate Q into Z-CNC coating instead of directly into PLA film? Why is better the development of bilayer structures instead a simpler monolayer PLA structure.

Line 210: luminosity should be changed by lightness

- The transparency of the films and the dispersion of CNC could not be evaluated from the lightness value which is a measure from the surface of the material. The authors are encouraged to provide the UV-visible spectra to evaluate the transparency and from it some insights concerning the CNC dispersion of the films can be performed (but not from lightness). Otherwise, please exclude lines 210-212 from the manuscript.

- It should be interesting to add photos of films to better understand the color results. From these photos some comments concerning the transparency can be also performed.

Figure 1. The quality of Figure 1 should be improved. Figure 1A unnecessary includes part of the spectra of Figure 1B.

- Line 244: which kind of chemical interactions take place between the antioxidant compound and cellulose nanocrystals?

- Lines 248-249: which kind of slight chemical interactions take place between zein and CNC? band from 1175 to 1167 cm-1 should be assigned.

- Line 250: Which groups of each component are involved in hydrogen bonding interactions?

- Did the author consider performing SEM to prove the good adhesion of the bilayer structure?

-Lines 254-257: The crystallinity could not be calculated in these formulations because authors did not known the proportion of PLA in the bilayer material. Thus, the crystallization of PLA can be better discussed in terms of the formation (or not) of disorder or order alpha crystals, or comparing the influence of each additive in both cold crystallization and melting enthalpy. But the calculated crystallinity is not real and it should be deleted from Table 2.

-Lines 262-263: What do authors mean whit superficial physical adhesion? Do the authors think that it should be related with the interfacial adhesion frequently discussed in composite materials.

-Lines 266-267: The melting temperature of PLA takes place between 150 and 180 ºC. Thus, it is not necessary to perform the DSC analysis to so high temperature of 250 ºC. Thus, degradation processes of Q and CNC should not take place in DSC (they should be studied by TGA). Please, avoid the conclusion iii and correct the temperature range in DSC method section of the manuscript.

Therefore, in Equation 2 the fraction of PLA component should be included (or avoid the equation as well as the calculation of the crystallinity).

- How was calculated the onset decomposition temperature (T10%, T5%?)? and what about the Volatile Release Rate? Please clarify these points in TGA method section.

-The reduction of the thermal stability of PLA due to zein based coating is evident from the reduction of the onset decomposition temperature. Nevertheless, the reduction of about 10 ºC in the maximum degradation temperature can be suggesting positive interaction between two polymeric matrices.

The first time that abbreviation is mentioned the meaning should be added (i.e. DTGA).

- Lines 286-288: an appropriate reference should be added.

- Lines 293-294: From Table 3 it seems that the thermal stability of the films did not improve by the incorporation of CNC and Q compared to the PLA/ZN control. Please, check these values or avoid this comment as well as those from lines 302-304.

- Line 311: These changes represented a relevant... should be changed by: These changes represent a relevant.

- Lines 314-315: the reduction of elongation at break accompanied by a reduction of both Young modulus and strength suggest a typical plasticizing effect, clearly produced due to the presence of zein coating in PLA matrix.

- Line 328: The sorption of water increased mobility of polymeric chains... should be changed by: The sorption of water increased the mobility of polymeric chains

- Line 337: "Permeability values of PLA is relatively low when compared to other bioplastics"... Could authors add some examples with the corresponding references?

- Release Kinetic of active component from the bilayer materials: it should be interesting to compare these results with already reported antioxidant PLA formulations with the objective to highlight (and also justify) the interest of adding the active component in the zein layer instead of directly into the PLA matrix.

References should be updated due to only 4 references (from 51) are directly related with bilayer or multilayer packaging formulations which is the main topic of the manuscript.

Author Response

The authors appreciate the positive evaluation of the reviewer. The proposed changes and modifications were carried out to improve the manuscript. The introduction was definitely improved.

Comments and Suggestions for Authors

Dr. Velasquez and co-workers report a very interesting work of using quercetin (Q) as active component of zein (Z)-cellulose nanocrystals (CNC) as coating for poly(lactic acid) (PLA) film. The obtained bilayer systems are intended for active food packaging. The topic is interest ad well related with the scope of the journal. The work is well organized and the materials were fully characterized, while the understanding of release process is very interesting. Therefore, I would like to recommend this work to be published in Polymers after minor revision to address some of my comments below:

-Line 31: poly (lactic acid) (PLA) should be poly(lactic acid) (PLA) (the space in polymer names should be avoided).

Corrections were carried out over the whole manuscript.

-Lines 37-39: Please add some references.

Two new references were included.

-Lines 55-56: some more recent references than 2014 from the last years (last two or three years) should be included.

References were changed to more recent references.

- Lines 64-64: some references should be added and discussed to justify the interest as well as the recent novelty of the topic suggested by the authors.

Introduction was carefully revised and improved, and several references were included.

- I suggest authors to describe better which is the aim of the work. Why authors decided to incorporate Q into Z-CNC coating instead of directly into PLA film? Why is better the development of bilayer structures instead a simpler monolayer PLA structure.

The incorporation of quercetin in a separate layer is preferred because it would be considered as the internal layer in the application of the material such as in the manufacture of packaging, where the release of the antioxidant is required only in the food contact area, minimizing losses to the environment or surroundings. The amount of quercetin necessary to develop this thin layer of about 8-10 μm is much lower (approx.. 10 times lower) than developing a whole active system with 100 μm.

In addition, there may be an increased risk of loss of quercetin during extrusion of the PLA between 200-210 °C, considering that quercetin can lose molecular water above 100 °C, as verified by TGA analysis for anhydrous quercetin (Figure 2). Furthermore, quercetin is a model active compound and, in the case of another active component with lower thermal stability, this system allows an effective protection of these compounds. Thus, zein is used as a compound carrier matrix and besides, this layer does not alter the biodegradability of the material. The selection of this biopolymer was due to its compatibility with quercetin and cellulose nanocrystals that were incorporated to study their positive effect on the release of the antioxidant. The coating technique allows a less aggressive application and drying with heat exposure in a short time.

In order to give importance of this system, these arguments were included in Introduction and principally regarding to quercetin release kinetics.

Line 210: luminosity should be changed by lightness.

Changed.

- The transparency of the films and the dispersion of CNC could not be evaluated from the lightness value which is a measure from the surface of the material. The authors are encouraged to provide the UV-visible spectra to evaluate the transparency and from it some insights concerning the CNC dispersion of the films can be performed (but not from lightness). Otherwise, please exclude lines 210-212 from the manuscript.

Authors agree with reviewer. According to the comments, the respective changes over section 3.1. were carried out in the revised manuscript.

- It should be interesting to add photos of films to better understand the color results. From these photos some comments concerning the transparency can be also performed.

Figure 1. The quality of Figure 1 should be improved. Figure 1A unnecessary includes part of the spectra of Figure 1B.

Figure 1 was modified and a new version with better quality (Figure 1_revised) was included in the revision. Regarding to the photos of the films, the authors believe that the photos will not provide much information since the differences on transparency between the developed bilayer films were not significant.

- Line 244: which kind of chemical interactions take place between the antioxidant compound and cellulose nanocrystals?

Antioxidant compound and cellulose nanocrystals presented chemical interactions mainly through hydrogen bonding-type interactions. FTIR spectra discussion was revised and short comments were included to clarify the argumentation in the revised manuscript (line 265).

Furthermore, it was indicated that peak at 1199 cm-1 corresponds to C-O stretching in phenol (Line 261 revised manuscript) (Catauro, M., Papale, F., Bollino, F., Piccolella, S., Marciano, S., Nocera, P., & Pacifico, S. (2015). Silica/quercetin sol–gel hybrids as antioxidant dental implant materials. Science and technology of advanced materials16(3), 035001).

Note: Authors do not think this reference needs to be included.

- Lines 248-249: which kind of slight chemical interactions take place between zein and CNC? band from 1175 to 1167 cm-1 should be assigned.

Hydrogen bonding-type interactions mainly between carbonyl and –NH groups from the zein and cellulose nanocrystals hydroxyl groups. Information about this band was included in the original manuscript. Band at 1175 cm-1 was related to C-N stretching of amine groups with its corresponding reference (Ref. 39), and it was included in FTIR discussion to clarify the explanation.

- Line 250: Which groups of each component are involved in hydrogen bonding interactions?

Hydrogen bonding interactions could be formed by i) quercetin hydroxyl and carbonyl groups, ii) cellulose hydroxyl groups and iii) –NH (amine) and carbonyl groups from the zein, but also oxygens of quercetin and CNC could be involved. The types of interactions were mentioned and related with some references in the original manuscript (Lines 299-303) but it was explained more clearly in the revised manuscript (Lines 330-333).

- Did the author consider performing SEM to prove the good adhesion of the bilayer structure?

The authors did not consider performing SEM microscopy because previous tests have evidenced the difficulty of confirming the good adhesion.

-Lines 254-257: The crystallinity could not be calculated in these formulations because authors did not known the proportion of PLA in the bilayer material. Thus, the crystallization of PLA can be better discussed in terms of the formation (or not) of disorder or order alpha crystals, or comparing the influence of each additive in both cold crystallization and melting enthalpy. But the calculated crystallinity is not real and it should be deleted from Table 2.

Corrected. The column of crystallinity values in Table 2 was removed and the respective modifications in the associated text was done. Regarding the enthalpies of cold crystallization and fusion, it is worthy to mention thermal parameters did not present statistically significant differences with respect to the PLA film, as specified in the legend of Table 2.

-Lines 262-263: What do authors mean whit superficial physical adhesion? Do the authors think that it should be related with the interfacial adhesion frequently discussed in composite materials.

Authors agree with reviewer. According to the suggestion, the word "superficial" was changed to "interfacial" on Line 291 of the revised manuscript.

-Lines 266-267: The melting temperature of PLA takes place between 150 and 180 ºC. Thus, it is not necessary to perform the DSC analysis to so high temperature of 250 ºC. Thus, degradation processes of Q and CNC should not take place in DSC (they should be studied by TGA). Please, avoid the conclusion iii and correct the temperature range in DSC method section of the manuscript.

Therefore, in Equation 2 the fraction of PLA component should be included (or avoid the equation as well as the calculation of the crystallinity).

According to the suggestion:

i) Corrections were carried out by deleting conclusion iii and their corresponding references. ii) The temperature range was corrected in the DSC method of the experimental section.

iii) Equation 2 and the corresponding text regarding the calculation of crystallinity were eliminated, which also implied the elimination of Reference 22.

- How was calculated the onset decomposition temperature (T10%, T5%?)? and what about the Volatile Release Rate? Please clarify these points in TGA method section.

These items were clarified in the experimental section 2.4.2.

-The reduction of the thermal stability of PLA due to zein based coating is evident from the reduction of the onset decomposition temperature. Nevertheless, the reduction of about 10 ºC in the maximum degradation temperature can be suggesting positive interaction between two polymeric matrices.

The authors agree on the reviewer's first assumption and even include this idea in the original version related to this early degradation. Nevertheless, they do not completely agree with the second assumption, since a positive interaction would increase thermal stability, and it was not the case.

- The first time that abbreviation is mentioned the meaning should be added (i.e. DTGA).

Corrected, meaning was included.

- Lines 286-288: an appropriate reference should be added.

References regarding the degradation of quercetin and CNC (Ref. 37 and 38) were included.

- Lines 293-294: From Table 3 it seems that the thermal stability of the films did not improve by the incorporation of CNC and Q compared to the PLA/ZN control. Please, check these values or avoid this comment as well as those from lines 302-304.

Authors agree with reviewer and modifications were carried out. Results evidenced the incorporation of CNC and Q had no effect on Tonset and Td, max temperatures compared to PLA/ZN control values. Text was corrected. Additionally, text and corresponding references were deleted.

- Line 311: These changes represented a relevant... should be changed by: These changes represent a relevant.

Corrected.

- Lines 314-315: the reduction of elongation at break accompanied by a reduction of both Young modulus and strength suggest a typical plasticizing effect, clearly produced due to the presence of zein coating in PLA matrix.

Authors agree with the plasticizing effect produced by the incorporation of zein coating in PLA matrix. Although this fact was already declared, a short sentence was included to clarify it.

The lines that reviewer has mentioned explain the influence of the incorporation of the antioxidant and cellulose nanocrystals in the bilayer structures. The results showed a reduction in Young's modulus and tensile strength of bilayer materials compared to PLA/ZN, whose differences were accentuated for the bilayer PLA/ZN.Q.CNC1 composite. Elongation at break values were increased. In this case, more than a plasticizing effect of zein, the interactions between the components of the coating and at the interface of the bilayer would promote the formation of a network with elastic capacity before the effort.

- Line 328: The sorption of water increased mobility of polymeric chains... should be changed by: The sorption of water increased the mobility of polymeric chains

Changed.

- Line 337: "Permeability values of PLA is relatively low when compared to other bioplastics"... Could authors add some examples with the corresponding references?

PLA polymer presents a lower water vapor transmission rate than other biodegradable polymers, such as polycaprolactone, cellulose acetate and cellulose acetate propionate. A reference was included.

- Release Kinetic of active component from the bilayer materials: it should be interesting to compare these results with already reported antioxidant PLA formulations with the objective to highlight (and also justify) the interest of adding the active component in the zein layer instead of directly into the PLA matrix.

In order to justify the interest of adding quercetin into the zein layer instead directly in the PLA matrix, these discussion of results was improved. Diffusion coefficients of quercetin of bilayer structures ranged between 1.5 and 4.5 x 10-15 m2 s-1 and these values were lower than the diffusion coefficient value reported for quercetin in PLA (6.4 x 10-15 m2 s-1), revealing that the inclusion of quercetin inside a zein layer coated PLA allows a more sustained release than could be obtained including quercetin in PLA structure. These results are very promising for the development of active food packaging materials of sustained release.

References should be updated due to only 4 references (from 51) are directly related with bilayer or multilayer packaging formulations which is the main topic of the manuscript.

It is not possible to add new references related to migration of quercetin from coated polymers, bilayer or multilayer formulations for food packaging because there are not more studies dealing with that.

Round 2

Reviewer 1 Report

I read carefully the revised version of the paper and the detailed answers to the reviewers.

The authors achieved an impressive work to improve the paper. The technical quality is obviously high and this paper deserves to be published.

Obviously, the material descirbed in the paper is new and the measured properties are interesting enough to be published.

When I read the first version, I had a main comment regarding the originality. Regarding the updated version and the reply to the reviewers,  I appreciate the update of the introduction on the multilayered materials. I agree that the work is original in the field of packaging.

Nevertheless, in the field of polymers, I see less originality, especially if I consider the components of the packaging materials (th single layers and thus the polymer). My point of view is that originality is little bit thin in the field of Polymers and thus for publication in a journal such as Polymers.

Based on these expactations, I consider that this paper should be published in a journal dealing specifically with packaging. Obviously, the other reviewer has a different point of view. This is a typical situation where the editor should have the final decision. I have thus decided to recommend to reject the paper to give the hand to the editor. Whatever the final decision of the editor, I can follow it due the technical quality of the paper.

Author Response

The authors respect the reviewer's opinion. However, taking the same words from the reviewer,authors believe that the technical quality of the work presented, the novelty, the interest of the measured properties and the originality are sufficient reasons for the publication of this work.

Many works published in Polymers are based on development of similar materials focused on biodegradable polymer for food packaging applications. The development and characterization of materials aimed at food packaging are of great interest, and authors do not understand the fact that it is not an important work.